# Documenting Pharmacogenomic Test Results in Electronic Health Records: Practical Considerations for Primary Care Teams

**DOI:** 10.3390/jpm11121296

**Published:** 2021-12-04

**Authors:** Roseann S. Gammal, Lucas A. Berenbrok, Philip E. Empey, Mylynda B. Massart

**Affiliations:** 1Department of Pharmacy Practice, Massachusetts College of Pharmacy and Health Sciences, Boston, MA 02115, USA; roseann.gammal@mcphs.edu; 2Department of Pharmacy and Therapeutics, University of Pittsburgh School of Pharmacy, Pittsburgh, PA 15261, USA; berenbrok@pitt.edu (L.A.B.); pempey@pitt.edu (P.E.E.); 3Department of Family Medicine, University of Pittsburgh School of Medicine, Pittsburgh, PA 15261, USA

**Keywords:** pharmacogenomics, pharmacogenetics, primary care, family medicine, electronic health record

## Abstract

With increasing patient interest in and access to pharmacogenomic testing, clinicians practicing in primary care are more likely than ever to encounter a patient seeking or presenting with pharmacogenomic test results. Gene-based prescribing recommendations are available to healthcare providers through Food and Drug Administration-approved drug labeling and Clinical Pharmacogenetics Implementation Consortium guidelines. Given the lifelong utility of pharmacogenomic test results to optimize pharmacotherapy for commonly prescribed medications, appropriate documentation of these results in a patient’s electronic health record (EHR) is essential. The current “gold standard” for pharmacogenomics implementation includes entering pharmacogenomic test results into EHRs as discrete results with associated clinical decision support (CDS) alerts that will fire at the point of prescribing, similar to drug allergy alerts. However, such infrastructure is limited to the few institutions that have invested in the resources and personnel to develop and maintain it. For the majority of clinicians who do not practice at an institution with a dedicated clinical pharmacogenomics team and integrated pharmacogenomics CDS in the EHR, this report provides practical tips for documenting pharmacogenomic test results in the problem list and allergy field to maximize the visibility and utility of results over time, especially when such results could prevent the occurrence of serious adverse drug reactions or predict therapeutic failure.

With increasing patient interest in and access to pharmacogenomic testing, clinicians practicing in primary care are more likely than ever to encounter a patient seeking or presenting with pharmacogenomic test results [1]. As costs for genetic testing lessen and insurance coverage expands, patients have fewer barriers to independently pursue pharmacogenomic testing [2]. Direct-to-consumer testing, clinical laboratories that employ physicians to place test orders on a patient’s behalf, or self-referral to a local practice offering clinical pharmacogenomics services are examples of pharmacogenomic testing services with little to no barriers for entry [3,4,5]. Federally-funded research projects, including the *All of Us* Research Project [6], and large-scale clinical implementations in the Veterans Affairs healthcare system [7] and other institutions [8] also provide avenues for pharmacogenomic testing for patients. Furthermore, patients are frequently prescribed medications with pharmacogenomic implications, and using pharmacogenomic test results to inform prescribing decisions can minimize trial and error processes and improve medication effectiveness and safety [9]. Recent studies have estimated that 58% of English primary care patients are prescribed at least one medication with an actionable pharmacogenomic variant [10]. In a population of U.S. veterans, 99% were estimated to be carriers of at least one actionable pharmacogenomic variant [11]. An “actionable” pharmacogenomic result is one that is associated with a change in medication selection, dosing, or monitoring due to the potential for an adverse drug reaction or therapeutic failure with standard use. Drug selection and dosing recommendations based on pharmacogenomic data are available through U.S. Food and Drug Administration (FDA)-approved labeling and the Clinical Pharmacogenetics Implementation Consortium (CPIC) guidelines (Table 1) [12,13]. Over 350 medications currently have pharmacogenomic information in their FDA-approved labeling, and there have been 26 CPIC guidelines published to date involving 23 genes and over 65 drugs, with new guidelines published each year [12,14].

Given the lifelong utility of pharmacogenomic test results to optimize pharmacotherapy for medications commonly prescribed in a primary care practice (e.g., analgesics, antidepressants, statins, anticoagulants, and proton pump inhibitors) and across specialty areas (e.g., infectious disease, oncology, and pediatrics), documenting pharmacogenomic test results in a patient’s electronic health record (EHR) is essential for continued use over time [15]. Large academic medical centers that have implemented system-wide pharmacogenomics clinical services integrate pharmacogenomic test results into EHRs in a discrete manner with accompanying clinical decision support (CDS) alerts for results that warrant a change from normal prescribing. This means that genotype (e.g., *CYP2C19 *1/*2*) and predicted phenotype (e.g., CYP2C19 intermediate metabolizer) information is documented as specific data fields, similar to the way a blood glucose result is entered, rather than simply scanning a document into an EHR. Discrete data entry then enables linking CDS to these specific actionable results, which is considered the “gold standard” for pharmacogenomics implementation [16,17,18,19]. Such an approach ensures that clinicians are alerted to both the availability of relevant pharmacogenomic test results and specific recommendations on how to best utilize available results at the point of prescribing. This approach is particularly important when multi-gene panels or preemptive testing is ordered, as some of the results may be immediately actionable, while others have long-term clinical utility or utility that has yet to be proven or discovered. In this scenario, the prescribing clinician may not be the one who initially ordered the test, but the pharmacogenomic result could have profound implications for future pharmacotherapy outcomes. In some cases, there could also be liability concerns if the pharmacogenomic information is not utilized and the patient experiences a severe adverse medication-related outcome that could have been prevented had the genetic information been taken into account per the latest guidelines. Appropriate documentation is therefore essential to drive gene-drug interaction alerts at the time of prescribing and to increase the likelihood that clinically relevant pharmacogenomic test results are used appropriately to optimize drug selection and dosing. Discrete results are also essential for research on EHR data, and CDS will also be necessary to identify gene–drug–environment interactions in the future. The problem and proposed solutions presented hereafter are primarily applicable to hospitals, clinics, and pharmacies without integrated pharmacogenomics CDS, though some of the discussed solutions may also be beneficial within health systems with full integration.

## 1. The Problem

Integrating pharmacogenomics into EHRs with customized CDS requires significant resources and specifically trained personnel to implement and maintain. As pharmacogenomic testing becomes more common, clinicians in primary care who work in a health system without integrated pharmacogenomics CDS need to determine how to best document pharmacogenomic results in EHRs to maximize the visibility and utility of results over time, especially when such results could prevent the occurrence of serious adverse drug reactions or predict therapeutic failure. In addition, clinicians need to account for the quality of the pharmacogenomic test results prior to documentation. Pharmacogenomic test results are provided to the primary care clinician from a variety of sources such as a testing laboratory or even by the patient themselves. First, a careful evaluation must be undertaken to understand the source of testing and whether it was FDA-approved or conducted in an accredited laboratory for clinical use (e.g., Clinical Laboratory Improvement Amendments (CLIA)-certified). Some direct-to-consumer genetic testing laboratories may have FDA clearance for clinical use (e.g., 23andMe’s *CYP2C19* test for clopidogrel and citalopram), but many do not (Figure 1) [20,21].

Currently, the standard practice upon receiving clinical pharmacogenomic test results may include uploading a scanned copy of the laboratory report (e.g., PDF file) into a patient’s EHR and/or documenting the results as free text in a clinical note. Storing information in paper-based formats at the “encounter level” is less than ideal because it lends itself to becoming difficult to find in a patient’s chart over time. Clinicians who may prescribe medications for said patient in the future will not necessarily know that pharmacogenomic data are available, in effect rendering the data useless for future prescribing. In addition, pharmacogenomics is a rapidly evolving field, and the information contained in a static report is not readily amenable to updates as new data emerges (e.g., new interpretation of genotype to phenotype, actionable gene/drug pairs, or gene-based prescribing recommendations). Pharmacogenomic data stored in a scanned report or clinical note text also cannot be linked to CDS should the health system decide to invest in developing informatics tools. Beyond considerations for a single health system, there is the significant barrier of the portability of patients’ pharmacogenomic results across institutions and EHRs, which limits the use of these important data.

Pharmacogenomic test results also require special considerations for their visibility and utility in a clinical electronic environment because they differ from many other types of genetic and laboratory results in several ways. First, because medications are metabolized by common pathways, one gene result could have clinical implications for multiple drugs, not just for the drug for which the pharmacogenomic test was initially ordered. Since few EHRs have a dedicated section for pharmacogenomics, the location of these results can vary and may not be easily discoverable. Second, the results need to be visible (not hidden) to all clinicians accessing the EHR, like a drug allergy. Third, as previously mentioned, pharmacogenomic test results have lifelong implications so need to be easily accessed perpetually and not archived. However, if a patient is genotyped for the same gene at multiple time points, there is a chance that the results could differ due to differences in the variants tested by the laboratory; this is in contrast to other lab tests (e.g., blood type and international normalized ratio), which yield the same results regardless of the laboratory that performs the test. Finally, the evolving evidence base could also result in the need for re-interpretation of the results over time as new evidence is assessed and included in the clinical guidelines.

## 2. Proposed Solutions

To mitigate these issues in a resource-limited setting (i.e., no lab interface for integrating discrete results with CDS alerts), one strategy that a clinician may consider is to enter a problem list entry for each actionable phenotype (e.g., CYP2D6 poor metabolizer) (Figure 1). These entries should be standardized phenotype terms established by CPIC that are available for use in some EHRs through the Systematized Nomenclature of Medicine–Clinical Terms (SNOMED-CT) [22]. Problem lists are frequently reviewed by clinicians; therefore, this strategy solves the aforementioned accessibility and visibility concerns. In addition, storing the results discretely in this manner may enable the development of gene–drug interaction alerts in the future.

At institutions where custom CDS is not feasible due to limited resources or trained personnel, primary care clinicians should also consider leveraging existing CDS drug allergy alerts via the allergy section of an EHR. While not ideal for all pharmacogenomic associations, this is strategic for actionable results that could have severe and potentially life-threatening consequences if ignored (e.g., an *HLA-B*15:02*-positive individual prescribed carbamazepine or a TPMT poor metabolizer prescribed standard doses of a thiopurine) (Figure 1). Drug entries in the allergy field, along with a brief comment about the relevant pharmacogenomic test result and its clinical consequence, will allow for active CDS alerts (in this case, “allergy” alerts) to fire if that medication is ever prescribed for the patient. This simple step is a proactive medication safety strategy to avoid serious and preventable adverse drug reactions. To avoid alert fatigue, it is recommended to only add pharmacogenomic test results to the allergy field if they could have potential serious and/or life-threatening consequences if ignored. Examples include medications for which pharmacogenomic data exist as an FDA boxed warning or contraindication (e.g., *HLA-B*/abacavir, *CYP2C19*/clopidogrel, and *G6PD*/rasburicase) [12].

A consideration of the aforementioned approaches involving problem list and allergy entries is that any clinician can edit these fields and/or remove information from them at any time, so education may be needed. In addition, if not enough supporting information is included with these entries, this could lead to confusion among clinicians and suboptimal prescribing decisions. When possible, context also should be provided to ensure clinician understanding of the gene–drug interaction and the appropriate prescribing recommendations, rather than simply listing the phenotype problem entry or medication as an “allergy” alone. 

To achieve pharmacogenomics results and CDS visibility beyond a single institution, data inter-operability and/or portability is needed. The U.S. healthcare system is notorious for its lack of inter-operability between institutions. While this is expected to improve due to new EHR data interoperability requirements advanced in the 2016 21st Century Cures Act, the current reality is that these data are rarely proactively distributed and accessed by downstream clinicians outside of the health systems or pharmacies where the data were originally documented [23,24]. Therefore, a patient may only benefit from pharmacogenomics CDS at the institution where it has been implemented, and should they receive care elsewhere, the results and associated CDS will not follow.

One solution to improve data portability involves educating patients to share their pharmacogenomic data and providing tools to facilitate this activity. Some leading pharmacogenomic testing institutions provide online data portals, wallet cards, or letters to lower barriers to this information dissemination [25]. For example, at St. Jude Children’s Research Hospital (St. Jude), where they have implemented a sophisticated, large-scale preemptive pharmacogenomic testing protocol, an important aspect of this implementation is to provide patients with an individualized letter describing their pharmacogenomic test results and the affected medications [19]. Patients are instructed to keep these letters and show them to their future providers outside of the St. Jude system. This use case illustrates the vital role of the clinician in empowering patients with knowledge about the clinical implications of their pharmacogenomic test results to optimize future medication therapy, as well as the limitations of data share between health systems. Educating patients about the utility of their pharmacogenomic test results and encouraging them to share the results with all future clinicians moving forward may be the best way to ensure their continued use over time, more so than customized pharmacogenomics EHR infrastructure that is restricted to a single location.

In parallel, broader pharmacogenomics education of clinicians would encourage routine documentation and integration of results when they exist. Simply asking patients about whether they have undergone pharmacogenomic testing previously is recommended to trigger such data collection. This is not unlike asking for current drug allergies and could become a core component of comprehensive medication management workflows. 

In conclusion, this report provides practical tips for primary care clinicians when documenting pharmacogenomic test results in EHRs. The workflow provided in Figure 1 provides a simple, stepwise process for documentation that is applicable in resource-limited settings to enable downstream use of these valuable data. We share these insights to maximize the utility of available pharmacogenomic test results to facilitate optimal drug selection and dosage and minimize the incidence of preventable adverse drug reactions at institutions without integrated pharmacogenomics CDS in EHRs. Most importantly, it is vital that clinicians educate patients about their pharmacogenomic test results and encourage them to share the results with their future clinicians.

## Figures and Tables

**Figure 1 jpm-11-01296-f001:**
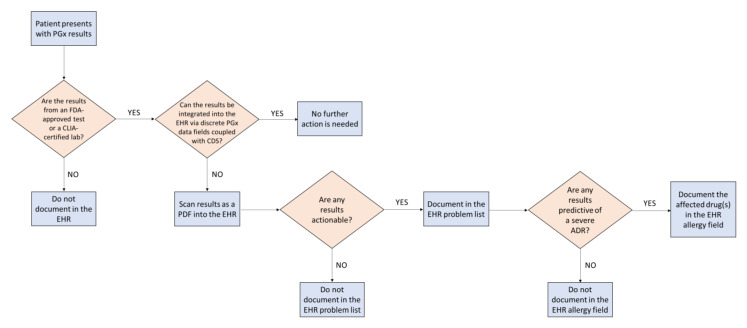
Decision-making process for documenting pharmacogenomic test results into electronic health records. ADR, adverse drug reaction; CDS, clinical decision support; CLIA, Clinical Laboratory Improvement Amendments; EHR, electronic health record; FDA, Food and Drug Administration; PGx, pharmacogenomics.

**Table 1 jpm-11-01296-t001:** Examples of drugs and associated genes with actionable prescribing recommendations from the Clinical Pharmacogenetics Implementation Consortium guidelines and/or U.S. Food and Drug Administration-approved drug labeling.

Medications	Genes
**Cardiology**	
Clopidogrel	*CYP2C19*
Simvastatin	*SLCO1B1*
Warfarin	*CYP2C9, CYP4F2, VKORC1*
**Gastroenterology**	
Dexlansoprazole	*CYP2C19*
Lansoprazole	*CYP2C19*
Metoclopramide	*CYP2D6*
Omeprazole	*CYP2C19*
Ondansetron	*CYP2D6*
Pantoprazole	*CYP2C19*
**Immunosuppressants**	
Azathioprine	*NUDT15, TPMT*
Mercaptopurine	*NUDT15, TPMT*
Tacrolimus	*CYP3A5*
**Infectious Disease**	
Abacavir	*HLA-B*
Atazanavir	*UGT1A1*
Efavirenz	*CYP2B6*
Voriconazole	*CYP2C19*
**Neurology**	
Carbamazepine	*HLA-A, HLA-B*
Oxcarbazepine	*HLA-B*
Phenytoin	*CYP2C9, HLA-B*
**Pain Management**	
Celecoxib	*CYP2C9*
Codeine	*CYP2D6*
Flurbiprofen	*CYP2C9*
Hydrocodone	*CYP2D6*
Ibuprofen	*CYP2C9*
Meloxicam	*CYP2C9*
Piroxicam	*CYP2C9*
Tramadol	*CYP2D6*
**Psychiatry**	
Amitriptyline	*CYP2D6, CYP2C19*
Aripiprazole	*CYP2D6*
Atomoxetine	*CYP2D6*
Brexpiprazole	*CYP2D6*
Citalopram	*CYP2C19*
Clobazam	*CYP2C19*
Clomipramine	*CYP2D6, CYP2C19*
Desipramine	*CYP2D6*
Doxepin	*CYP2D6, CYP2C19*
Escitalopram	*CYP2C19*
Fluvoxamine	*CYP2D6*
Imipramine	*CYP2D6, CYP2C19*
Nortriptyline	*CYP2D6*
Paroxetine	*CYP2D6*
Pimozide	*CYP2D6*
Sertraline	*CYP2C19*
Trimipramine	*CYP2D6, CYP2C19*
Vortioxetine	*CYP2D6*
**Urate-Lowering Therapy**	
Allopurinol	*HLA-B*
Rasburicase	*G6PD*

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
