# Peer review of "Documenting Pharmacogenomic Test Results in Electronic Health Records: Practical Considerations for Primary Care Teams"

_jpm, 2021, doi:10.3390/jpm11121296_

Round 1

Reviewer 1 Report

This commentary provides suggestions for documenting pharmacogenomic testing results in the EHR focusing on the primary care setting. The authors describe the increasing use of pharmacogenomic testing and need to ensure adequate documentation of results. They describe the current gold standard of incorporating pharmacogenomic results as discrete entries in the EHR and use of CDS to alert prescribers to gene-drug interactions at the point of prescribing. The authors note that many providers may see pharmacogenomic test results without the ability to have results entered discretely or employ pharmacogenomic specific CDS. They describe the importance of documenting pharmacogenomic results in a manner that allows the results to be in the foreground of the patient’s medical record so that results are not overlooked. The authors provider two suggestions for documenting pharmacogenomic results when discrete integration is not available. The first option is to incorporate actionable phenotypes on the problem list. The second option is to enter actionable results in the allergies list/tab, thus allowing allergy alerts to act in a similar manner as pharmacogenomic-specific CDS. Finally, the authors briefly discuss concerns related to result portability and offer a few examples of ways to overcome this issue.

Including some statistics or information about rates of drug prescribing for drugs with pharmacogenomic implications in primary care could better support the focus on primary care. The commentary offers a somewhat novel solution to a well-established concern in the field. However the article is fairly surface level and could offer more insight into how these solutions would realistically be utilized. Given that the target audience for this article seems to be those who are not well versed in pharmacogenomics, adding a definition of what constitutes an actionable phenotype would be helpful for the readers. It would also strengthen the article if the authors were to further describe or provide examples of how they suggest reporting actionable phenotypes on the problem list. The authors could consider providing a specific example gene that follows the through the thought process of the provided figure to show different scenarios of when or how the result could be incorporated into the EHR. The authors do provide some discussion of concerns for reporting actionable results through the allergies list, but it would be helpful to discuss some possible concerns or limitations for utilizing the problem list.

Author Response

RE: RESPONSES TO REVIEWERS’ COMMENTS ON MANUSCRIPT ID jpm-1464667

Dear Reviewer, 

Point1:

This commentary provides suggestions for documenting pharmacogenomic testing results in the EHR focusing on the primary care setting. The authors describe the increasing use of pharmacogenomic testing and need to ensure adequate documentation of results. They describe the current gold standard of incorporating pharmacogenomic results as discrete entries in the EHR and use of CDS to alert prescribers to gene-drug interactions at the point of prescribing. The authors note that many providers may see pharmacogenomic test results without the ability to have results entered discretely or employ pharmacogenomic specific CDS. They describe the importance of documenting pharmacogenomic results in a manner that allows the results to be in the foreground of the patient’s medical record so that results are not overlooked. The authors provider two suggestions for documenting pharmacogenomic results when discrete integration is not available. The first option is to incorporate actionable phenotypes on the problem list. The second option is to enter actionable results in the allergies list/tab, thus allowing allergy alerts to act in a similar manner as pharmacogenomic-specific CDS. Finally, the authors briefly discuss concerns related to result portability and offer a few examples of ways to overcome this issue.

Including some statistics or information about rates of drug prescribing for drugs with pharmacogenomic implications in primary care could better support the focus on primary care.

Thank you for the suggestion. We added the following statistics and cited supporting references about pharmacogenomics and primary care to the introduction on pgs. 1-2:

“Recent studies estimate that  58% of English primary care patients were prescribed at least one medication with an actionable pharmacogenomic variant.10 In a population of US veterans, 99% were estimated to be carriers of at least one actionable pharmacogenomic variant.11

Point 2:

The commentary offers a somewhat novel solution to a well-established concern in the field. However, the article is fairly surface level and could offer more insight into how these solutions would realistically be utilized.

Thank you for this feedback.  Our goal with this article is to raise awareness and provide simple solutions applicable for primary care providers in the broadest of settings.  Significant additional depth is unfortunately beyond scope due to the word limitations of a brief commentary article.

Point 3:

Given that the target audience for this article seems to be those who are not well versed in pharmacogenomics, adding a definition of what constitutes an actionable phenotype would be helpful for the readers.

We added the following definition to pg. 2: “An “actionable” pharmacogenomic result is one that is associated with a change in medication selection, dosing, or monitoring due to the potential for an adverse drug reaction or therapeutic failure with standard use.”

Point 4: 

It would also strengthen the article if the authors were to further describe or provide examples of how they suggest reporting actionable phenotypes on the problem list. The authors could consider providing a specific example gene that follows the through the thought process of the provided figure to show different scenarios of when or how the result could be incorporated into the EHR.

Examples of how to report actionable phenotypes as well as the allergy field are provided.  RE: the problem list, please see  pg. 5: “To mitigate these issues in a resource-limited setting (i.e., no lab interface for integrating discrete results with CDS alerts), one strategy that a clinician may consider is to enter a problem list entry for each actionable phenotype (e.g., CYP2D6 poor metabolizer). These entries should be standardized phenotype terms established by CPIC which are available for use in some EHRs through the Systematized Nomenclature of Medicine – Clinical Terms (SNOMED-CT).”

In addition, specific examples are provided for results that may be best suited for the allergy field on pg. 5: “At institutions where custom CDS is not feasible due to limited resources or trained personnel, primary care clinicians may also consider leveraging existing CDS allergy alerts via the allergy section of the EHR. While not ideal for all pharmacogenomic associations, this is strategic for actionable results that could have severe and potentially life-threatening consequences if ignored (e.g., an HLA-B*15:02 positive individual prescribed carbamazepine or a TPMT poor metabolizer prescribed standard doses of a thiopurine).”

Point 5:

The authors do provide some discussion of concerns for reporting actionable results through the allergies list, but it would be helpful to discuss some possible concerns or limitations for utilizing the problem list.

Thank you for this suggestion. We added the following paragraph to pg. 5:

“A consideration of the aforementioned approaches involving problem list and allergy entries is that any clinician can edit these fields and/or remove information from them at any time so education may be needed. In addition, if not enough supporting information is included with these entries, this could lead to confusion among clinicians and suboptimal prescribing decisions. When possible, context also should be provided in to ensure clinician understanding of the gene-drug interaction and the appropriate prescribing recommendations, rather than simply listing the phenotype problem list entry or medication as an “allergy” alone.”

Reviewer 2 Report

Dr. Roseann Gammal and authors highlights strategies to incorporate pharmacogenomic test results in the electronic health records specific to primary care providers that may not have formal clinical decision support for pharmacogenomic results. The needs and special considerations outlined in this article is timely as accessibility of pharmacogenomic testing grows and providers are inevitably approached by their patients request either obtain pharmacogenomic testing or interpreting results.

The relevance of this article is important as access to pharmacogenomic testing grows with costs lessening, expanding Medicare local coverage in 40 out of the 50 states and commercial payers adopting coverage, and availability growing with offerings from direct-to-consumer genetic companies, nationwide laboratory networks, commercial genetic companies, and institutional pathology or molecular diagnostic lab. Despite this rise, unlike large academic health systems that have formally implemented pharmacogenomics, primary care clinic may not have the same level of resources to formally incorporate and document results so that they maintain their utility years later. This article is well-written and I anticipate it will be well-received. 

The article flows nicely from the introduction/background, problem of documentation, and proposed solution. Dr. Roseann Gammal, et al. clearly identifies methods that is commonly used in any electronic health record (such as documenting in the problems list or using allergy fields) and allows time-independent documentation with minimal needs for informatic resources. Table 1 and Figure 1 are simple and easy to follow.

I do not have any suggestions for this article. The majority of cited references are recent and appropriately supports the statements and conclusions outlined in this article. I have not found any recent publications specific to this topic. No inaccuracies within the text or sentences identified.

Author Response

thank you for your review

Reviewer 3 Report

Background provides a good description of why it is important to have PGx results available in the EHR as discrete data (e.g., so results are readily accessible and visualizable to clinicians, can be updated over time, and can be linked to CDS alerts). It also appropriately highlights the problem that most medical centers do not have the resources/EHR infrastructure built to accommodate PGx results as discrete data. Figure 1 is very useful.

Major

The article clearly outlines the problem that a lack of structured data imposes and it provides a clear solution to generate structured data. However, the article could benefit from expanded examples of what can be done with this structured data. As the authors mention, an active CDS alert is certainly a common solution. But are there other strategies that these structured data could be used to help inform care?

Minor

The abstract describes the rationale for the paper but not discuss solutions to the problems raised. Suggest including a summary of the proposed solutions in the abstract.

What are the limitations of the proposed approach of utilizing a problem list and allergy list? For example, these fields are editable and the medication may be removed. I agree with authors that listing the phenotype is a good idea expand on the limitations of listing specific medications (e.g., “patient says I’m not allergic to codeine”).

Page 4, line 92 addresses that a careful evaluation needs to be completed of the PGx test used before adding results to the EHR. Consider adding a comment about assessing whether the test used covers appropriate variants for each gene.

Request the authors expand on the definition of “accredited laboratory with sufficient approvals for clinical use.” Especially in light of 23andMe having FDA clearance to offer PGx testing.

Please define “actionable result.”

Page 5, line 154 introduces the problem of lack of data inter-operability/portability between different institutions. This is brought up for the first time in the solutions section. Recommend introducing the problem earlier in the paper under problems section for consistency

This manuscript is focused on the clinicians who work in a health-system without integrated PGx CDS. Would call that out more clearly. Would any of this be applicable to those who work in a health system with integrated PGx CDS?

Author Response

RE: RESPONSES TO REVIEWERS’ COMMENTS ON MANUSCRIPT ID jpm-1464667

Dear Reviewer:

Point 1: 

Background provides a good description of why it is important to have PGx results available in the EHR as discrete data (e.g., so results are readily accessible and visualizable to clinicians, can be updated over time, and can be linked to CDS alerts). It also appropriately highlights the problem that most medical centers do not have the resources/EHR infrastructure built to accommodate PGx results as discrete data. Figure 1 is very useful.

Thank you.

Point 2: 

Major

The article clearly outlines the problem that a lack of structured data imposes and it provides a clear solution to generate structured data. However, the article could benefit from expanded examples of what can be done with this structured data. As the authors mention, an active CDS alert is certainly a common solution. But are there other strategies that these structured data could be used to help inform care?

Thank you,

article currently states: Such an approach ensures that clinicians are alerted to both the availability of relevant pharmacogenomic test results and specific recommendations on how to best utilize available results at the point of prescribing.  This approach is particularly important when multi-gene panels or preemptive testing is ordered as some of the results may be immediately actionable, while others have long-term clinical utility or utility that has yet to be proven or discovered. In this scenario, the prescribing clinician may not be the one who initially ordered the test, but the pharmacogenomic result could have profound implications for future pharmacotherapy outcomes. In some cases, there also could be liability concerns if the pharmacogenomic information was not utilized and the patient experienced a severe adverse medication-related outcome that could have been prevented had the genetic information been taken into account per the latest guidelines. Appropriate documentation is therefore essential to drive gene-drug interaction alerts at the time of prescribing and increase the likelihood that clinically relevant pharmacogenomic test results are used appropriately to optimize drug selection and dosing.

The following has been added to line 87.

Discrete results are also essential for research on EHR data, and CDS will also be necessary to identify gene-drug-environment interactions in the future.

Point 3:

Minor

The abstract describes the rationale for the paper but not discuss solutions to the problems raised. Suggest including a summary of the proposed solutions in the abstract.

Thank you for this suggestion. We edited the abstract to specify documenting pharmacogenomic test results in the allergy field and problem list, which are our proposed solutions when more discrete documentation systems do not exist.

Point 4:

What are the limitations of the proposed approach of utilizing a problem list and allergy list? For example, these fields are editable and the medication may be removed. I agree with authors that listing the phenotype is a good idea expand on the limitations of listing specific medications (e.g., “patient says I’m not allergic to codeine”).

Thank you for this suggestion. We added the following paragraph to pg. 5:

“A consideration of the aforementioned approaches involving problem list and allergy entries is that any clinician can edit these fields and/or remove information from them at any time so education may be needed. In addition, if not enough supporting information is included with these entries, this could lead to confusion among clinicians and suboptimal prescribing decisions. When possible, context also should be provided in to ensure clinician understanding of the gene-drug interaction and the appropriate prescribing recommendations, rather than simply listing the phenotype problem list entry or medication as an “allergy” alone.”

Point 5:

Page 4, line 92 addresses that a careful evaluation needs to be completed of the PGx test used before adding results to the EHR. Consider adding a comment about assessing whether the test used covers appropriate variants for each gene.

Although assessing variant coverage is an important consideration for choosing a pharmacogenomic test for a patient, if a patient presents with a pharmacogenomic test from a CLIA-certified clinical lab and it is used in decision making, that result should be entered in the electronic health record even if the variant coverage is suboptimal, especially if clinically relevant variants were detected. 

Point 6: 

Request the authors expand on the definition of “accredited laboratory with sufficient approvals for clinical use.” Especially in light of 23andMe having FDA clearance to offer PGx testing.

Thank you for this suggestion. We added in a specific comment about this on pg. 4, along with a new supporting citation (ref 21):

“First, a careful evaluation must be undertaken to understand the source of testing and whether it was FDA-approved or conducted in an accredited laboratory for clinical use (e.g., Clinical Laboratory Improvement Amendments (CLIA)-certified). Some direct-to-consumer genetic testing laboratories may have FDA clearance for clinical use (e.g., 23andMe’s CYP2C19 test for clopidogrel and citalopram), but many do not (Figure 1).”

Point 7:

Please define “actionable result.”

We added the following definition to pg. 2: “An “actionable” pharmacogenomic result is one that is associated with a change in medication selection, dosing, or monitoring due to the potential for an adverse drug reaction or therapeutic failure with standard use.”

Point 8:

Page 5, line 154 introduces the problem of lack of data inter-operability/portability between different institutions. This is brought up for the first time in the solutions section. Recommend introducing the problem earlier in the paper under problems section for consistency

Thank you for this suggestion. On pg. 4-5, we added the following sentence in the Problems section: “Beyond considerations for a single health-system, there is the significant barrier of portability of patients’ pharmacogenomic results across institutions and EHRs, which limits use of these important data.”

Point 9:

This manuscript is focused on the clinicians who work in a health-system without integrated PGx CDS. Would call that out more clearly. Would any of this be applicable to those who work in a health system with integrated PGx CDS?

The following sentence was added to pg. 3: “The problem and proposed solutions presented hereafter are primarily applicable to hospitals, clinics, and pharmacies without integrated pharmacogenomic CDS, though some solutions discussed may be beneficial even within health-systems with full integration.”